# Predicting suicide attempt or suicide death following a visit to psychiatric specialty care: A machine learning study using Swedish national registry data

Qi Chen[1]*, Yanli Zhang-James[2], Eric J. Barnett[3,4], Paul Lichtenstein[1], Jussi Jokinen[5,6], Brian M. D'Onofrio[1,7], Stephen V. Faraone[2,3], Henrik Larsson[1,8], Seena Fazel[9]

1 Department of Medical Epidemiology and Biostatistics, Karolinska Institutet, Stockholm, Sweden, 2 Department of Psychiatry and Behavioral Sciences, SUNY Upstate Medical University, Syracuse, New York, United States of America, 3 Department of Neuroscience and Physiology, SUNY Upstate Medical University, Syracuse, New York, United States of America, 4 College of Medicine, MD Program, SUNY Upstate Medical University, Syracuse, New York, United States of America, 5 Department of Clinical Neuroscience, Karolinska Institutet, Stockholm, Sweden, 6 Department of Clinical Sciences/Psychiatry, Umeå University, Umeå, Sweden, 7 Department of Psychological and Brain Sciences, Indiana University, Bloomington, Indiana, United States of America, 8 School of Medical Sciences, Örebro University, Örebro, Sweden, 9 Department of Psychiatry, Warneford Hospital, University of Oxford, Oxford, United Kingdom

* qi.chen@ki.se

**Data Availability Statement:** The data underlying this study contain sensitive personal information and therefore cannot be made freely available as

## Abstract

### Background

Suicide is a major public health concern globally. Accurately predicting suicidal behavior remains challenging. This study aimed to use machine learning approaches to examine the potential of the Swedish national registry data for prediction of suicidal behavior.

### Methods and findings

The study sample consisted of 541,300 inpatient and outpatient visits by 126,205 Sweden-born patients (54% female and 46% male) aged 18 to 39 (mean age at the visit: 27.3) years to psychiatric specialty care in Sweden between January 1, 2011 and December 31, 2012. The most common psychiatric diagnoses at the visit were anxiety disorders (20.0%), major depressive disorder (16.9%), and substance use disorders (13.6%). A total of 425 candidate predictors covering demographic characteristics, socioeconomic status (SES), electronic medical records, criminality, as well as family history of disease and crime were extracted from the Swedish registry data. The sample was randomly split into an 80% training set containing 433,024 visits and a 20% test set containing 108,276 visits. Models were trained separately for suicide attempt/death within 90 and 30 days following a visit using multiple machine learning algorithms. Model discrimination and calibration were both evaluated. Among all eligible visits, 3.5% (18,682) were followed by a suicide attempt/death within 90 days and 1.7% (9,099) within 30 days. The final models were based on ensemble learning that combined predictions from elastic net penalized logistic regression, random forest, gradient boosting, and a neural network. The area under the receiver operating characteristic

they are subject to secrecy in accordance with the Swedish Public Access to Information and Secrecy Act. Data can be made available to researchers who apply for approval by the Swedish Central Ethical Review Board (kansli@cepn.se). Requests for data can be made to the Department of Medical Epidemiology and Biostatistics in Karolinska Institutet (internservice@meb.ki.se).

**Funding:** This study was funded by the Wellcome Trust (#202836/Z/16/Z). YZJ is supported by the European Union's Seventh Framework Programme for research, technological development and demonstration under grant agreement No 602805 and the European Union's Horizon 2020 research and innovation programme under grant agreements No 667302. SVF is supported by the European Union's Seventh Framework Programme for research, technological development and demonstration under grant agreement No 602805, the European Union's Horizon 2020 research and innovation programme under grant agreements No 667302 & 728018 and NIMH grants 5R01MH101519 and U01 MH109536-01; all outside the submitted work. The funders had no role in study design, data collection and analysis, decision to publish, or preparation of the manuscript.

**Competing interests:** I have read the journal's policy and the authors of this manuscript have the following competing interests: JJ participated in an Advisory Board for Janssen. In the past year, SVF received income, potential income, travel expenses continuing education support and/or research support from Vallon, Tris, Otsuka, Arbor, Ironshore, Shire, Akili Interactive Labs, VAYA, Ironshore, Sunovion, Supernus and Genomind. With his institution, he has US patent US20130217707 A1 for the use of sodium-hydrogen exchange inhibitors in the treatment of ADHD. HL has served as a speaker for Evolan Pharma and Shire/Takeda and has received research grants from Shire/Takeda; all outside the submitted work.

**Abbreviations:** ATC, anatomical therapeutic chemical; AUCs, area under the ROC curves; LISA, longitudinal integration database for health insurance and labor market studies; NPV, negative predictive value; PPV, positive predictive value; ROC, receiver operating characteristic; SES, socioeconomic status; TRIPOD, transparent reporting of a multivariable prediction model for individual prognosis or diagnosis.

(ROC) curves (AUCs) on the test set were 0.88 (95% confidence interval [CI] = 0.87–0.89) and 0.89 (95% CI = 0.88–0.90) for the outcome within 90 days and 30 days, respectively, both being significantly better than chance (i.e., AUC = 0.50) ($p < 0.01$). Sensitivity, specificity, and predictive values were reported at different risk thresholds. A limitation of our study is that our models have not yet been externally validated, and thus, the generalizability of the models to other populations remains unknown.

## Conclusions

By combining the ensemble method of multiple machine learning algorithms and high-quality data solely from the Swedish registers, we developed prognostic models to predict short-term suicide attempt/death with good discrimination and calibration. Whether novel predictors can improve predictive performance requires further investigation.

## Author summary

### Why was this study done?

- Suicidal behavior is overrepresented in people with mental illness and contributes to the substantial public health burden of psychiatric conditions. Accurately predicting suicidal behavior has long been challenging.

- The potential of applying machine learning to linked national datasets to predict suicidal behavior remains unknown.

### What did the researchers do and find?

- We identified a sample of 541,300 inpatient and outpatient visits to psychiatric specialty care in Sweden during 2011 and 2012. The sample was then divided into a training dataset and a test dataset.

- We first trained prediction models separately for suicide attempt/death within 90 days and 30 days following a visit to psychiatric specialty care, using 4 different machine learning algorithms. We then used an ensemble method to combine the performance of the trained models with the intention to achieve an overall performance superior than each individual model.

- The final model based on the ensemble method achieved the best predictive performance. This model was applied to test dataset and showed good model discrimination and calibration for both the 90-day and 30-day outcomes.

### What do these findings mean?

- Our findings suggest that combining machine learning with registry data has the potential to accurately predict short-term suicidal behavior.

- An approach combining 4 machine learning methods showed an overall predictive performance slightly better than each individual model.

## Introduction

Suicide is a major public health concern globally. Predicting suicidal behavior is challenging both at the population level and among high-risk groups. The accuracy of predicting suicidal behavior based on clinical judgment varies considerably across clinicians. Risk factors known to be strongly associated with suicidal behaviors are weak predictors on their own [1]. One meta-analysis assessing the sensitivity and specificity of 15 different instruments for suicide and suicide attempt concluded that none of these instruments provided sufficient diagnostic accuracy defined by the authors (i.e., 80% for sensitivity and 50% for specificity) [2]. However, using a lower threshold for discrimination measures, it is possible for a prediction model to achieve the specified diagnostic accuracy, though most likely at the cost of a reduced positive predictive value (PPV). Another meta-analysis assessing the performance of previously reported psychological scales, biological tests, and third-generation scales derived from statistical modeling (mostly using conventional multivariable regression) for prediction of suicidal behavior [3]. The authors, who did not synthesize performance metrics other than PPV, reported a pooled PPV of 39% for the third-generation scales for predicting suicide attempts/deaths. One potential explanation for the modest predictive performance is that the data used for previous model development did not contain enough information for making accurate prediction. It is also possible that prediction of suicidal behavior is too complex to be based on a few simplified theoretical hypotheses [4].

Although currently not feasible to implement, difficult to understand by clinicians, and lacking transparency [5], machine learning algorithms have been applied to large-scale data such as electronic medical records for predicting suicidal behavior. In machine learning analytics, selecting candidate predictors may benefit from established theories and clinical expertise. When given access to large amounts of new data, machine learning may serve as an efficient and flexible approach to exploring the predictive potential of the new data. Meanwhile, machine learning algorithms usually identify far more complex data patterns than conventional methods, though at the cost of decreased interpretability [4]. Belsher and colleagues systematically reviewed 64 machine learning–based prediction models for suicide and suicide attempts in 17 studies and found that, despite good overall discrimination accuracy, the PPVs remained low, with inadequate information on negative predictive value (NPV) and calibration [6]. In their subsequent simulation analyses, they demonstrated that the achievable PPV was limited by the rarity of suicide even when sensitivity and specificity are hypothetically set to be nearly perfect. They thus recommended future research focus on predicting more common outcomes such as suicide attempts [6].

Most prior studies on predicting suicide or suicide attempts have been limited by small sample sizes. Very few of them provided a comprehensive report of model discrimination, including sensitivity, specificity, PPV, and NPV, as well as calibration. Only a single type of model (e.g., random forest model or regression model) was selected in most studies. Data from multiple Swedish national registers have been used to develop a multivariable regression model for predicting suicide among patients with schizophrenia spectrum disorders and bipolar disorder [7]. To date, the data have never been combined with machine learning to predict suicidal behavior in the setting of psychiatric specialty care.

In this study, we aimed to examine the achievable performance of models trained by several machine learning algorithms using the Swedish registry data. We developed prognostic prediction models for suicide attempt/death within 90 and 30 days following an in-/outpatient visit to psychiatric specialty care, using predictors generated via linkage between multiple Swedish national registers.

## Methods

### Ethics approval

The study was approved by the Regional Ethical Review Board in Stockholm, Sweden (reference number: 2013/862-31/5). The requirement for informed consent was waived because of the retrospective nature of the registry-based study.

### Data sources

Each individual registered as resident in Sweden is assigned a unique personal identity number, enabling linkage between the Swedish national registers [8]. The registers used in the current study are listed as follows: The Medical Birth Register covers nearly all deliveries in Sweden since 1973 [9]; The Total Population Register was established in 1968 and contains data on sex, birth, death, migration, and family relationship for Swedish residents who were born since 1932 [10]; The Multi-Generation Register, as part of the Total Population Register, links individuals born since 1932 and registered as living in Sweden since 1961 to their biological parents [11]; The longitudinal integration database for health insurance and labor market studies (LISA) launched in 1990 and contains annually updated data on socioeconomic status (SES) such as education, civil status, unemployment, social benefits, family income, and many other variables for all Swedish residents aged 16 years or older [12]; The National Patient Register covers inpatient care since 1964 (psychiatric care since 1973) and outpatient visits to specialty care since 2001, with a PPV of 85% to 95% for most disorders [13]; The Prescribed Drug Register contains data on all prescribed medications dispensed at pharmacies in Sweden since July 2005 [14]; Active ingredients of medications are coded according to the anatomical therapeutic chemical (ATC) classification system [14]; The National Crime Register provides data on violent and nonviolent crime convictions since 1973 [15].

### Study sample

The study sample consisted of any in-/outpatient visits by a patient aged 18 to 39 to psychiatric specialty care in Sweden between January 1, 2011 and December 31, 2012, with a primary diagnosis of any mental and behavioral disorders according to the International Classification of Diagnosis, 10th edition (ICD-10: F00–F99). To ensure reasonable data quality and minimize missingness of the predictors, only Sweden-born patients were included in the study. Patients who emigrated before the visit died on the same day of the visit, or lacked information on identity of either parent were excluded. A flowchart for identification of the study sample can be found in the online supplement (S1 Fig). The final study sample included 541,300 eligible visits by 126,205 patients during the study period.

### Outcome

In the current study, the outcome of interest was suicidal event, either suicide attempt or death by suicide. Consistent with previous research [16], suicide attempt was defined as intentional self-harm (ICD-10: X60–X84) or self-harm of undetermined intent (ICD-10: Y10-Y34) in the National Patient Register. Only unplanned inpatient or outpatient visits with a recorded self-harm were labeled as incident suicide attempts. Planned visits were likely to be follow-up healthcare appointments following an incident suicide attempt and thus were not classified as suicide attempts for our analysis. Any hospitalization, including stay at emergency department, stretching over more than 1 day is only registered once (as it is based on a discharge date) and thus was coded as 1 visit. Suicide was defined as any recorded death from intentional self-harm (ICD-10: X60–X84) or self-harm of undetermined intent (ICD-10: Y10-Y34) in the

Cause of Death Register. We chose to predict suicide attempt/death within 2 time windows, namely 90 and 30 days following a visit to psychiatric specialty care, given the time windows are likely to be meaningful for short- to medium-term interventions. These 2 outcomes were selected also to ensure a certain proportion of positive cases (more than 1%) and to facilitate comparison with prior studies [16,17].

## Predictors

When selecting potential predictors, we took into consideration previous studies on suicidal behavior [7,16,18,19] and the availability and quality of the information in the Swedish national registers. Predictors prior to or at each visit covered demographic characteristics (sex and age at the visit), SES (family income, educational attainment, civil status, unemployment, and social benefits), electronic medical records (planned/unplanned visit, in-/outpatient visit, clinical diagnoses of psychiatric and somatic disorders, intentional self-harm and self-harm of undetermined intent, methods used for self-harm, and dispensed medications), criminality (violent and nonviolent criminal offenses), and family history of disease and crime. To better utilize information on timing of the predictors, we generated predictors related to clinical diagnoses using several arbitrary time windows (i.e., at the index visit, within 1 month, 1 to 3 months, 3 to 6 months, 6 to 12 months, 1 to 3 years, and 3 to 5 years before the index visit), assuming better predictive power by events occurring more recently. Prior intentional self-harm and self-harm of undetermined intent were treated as separate predictors. Methods for prior self-harm were first categorized according to the first 3 digits of the ICD-codes (ICD-10: X60–X84, Y10-Y34). Intentional self-poisoning (X60–X69) and self-poisoning of undetermined intent (Y10–Y19) were then combined into 2 separate predictors (S3 Table). Predictors related to dispensed medications within different time windows (i.e., within 1 month, 1 to 3 months, 3 to 6 months, 6 to 12 months, 1 to 3 years, and 3 to 5 years before the visit) were generated in the same way as for clinical diagnoses. The complete ICD and ATC codes used for ascertainment of clinical diagnoses and dispensed medications are listed in S2 and S4 Tables. Age at the visit and family income were treated as continuous variables and rescaled to the range of 0 to 1. The other predictors were treated as binary or categorical. Missing predictors appeared to co-occur in the same person. Therefore, no imputation was done as missing at random could not be assumed. Missingness on the predictors ranged from 0.6% to 12.5% (S1 Table) and was coded as a separate category for each predictor. All categorical predictors were converted to dummy indicators. In total, 425 predictors were included for the subsequent model derivation (S1 Table).

## Model derivation and evaluation

We treated visits by each patient as a separate cluster and randomly split the entire study sample by patient cluster into an 80% training set containing 433,024 visits by 100,964 patients and a 20% test set containing 108,276 visits by 25,241 patients. The purpose of splitting by patient cluster was to prevent a model from performing artificially well on the test set due to redundancy between the training and test sets.

We first trained 4 models using elastic net penalized logistic regression [20], random forest [21], gradient boosting [22], and a neural network [23]. The 4 model algorithms were selected, first, because they have been repeatedly used in previous research but have never been applied to the same data in the same study; second, because the models diverse in analytic approach, which makes it possible to be aggregated using an ensemble method to achieve a predictive performance better than each individual model. For each model, we grid-searched for the optimal set of hyperparameters via 10-fold cross-validation and used the area under the receiver

operating characteristic (ROC) curve (AUC) as the evaluation metric. We then compared the performance of the best models trained by the 4 algorithms, together with the ensemble models that used the average predicted risk of 2 or more of the best models for making prediction [24]. Based on the results of cross-validation, among the models giving the highest average validation AUC (values unchanged to the fourth decimal place after rounding were considered the same), the one showing the smallest difference between the training and validation AUCs was selected and applied to the entire training set to obtain the final model parameters. The test set was reserved only for the final model validation.

The AUC for the test set was used to evaluate model discrimination (i.e., the extent to which a prediction model can separate those who will experience suicidal events from those who will not). The confidence interval (CI) of the test AUC was estimated using Delong's method [25]. Additional metrics, including sensitivity, specificity, PPV, and NPV, were reported over a series of risk thresholds. Sensitivity measures the proportion of predicted positives among all true positives and specificity measures the proportion of predicted negatives among all true negatives [26]. These 2 metrics represent the characteristics of a prediction model, which are not affected by the prevalence of the predicted outcome. PPV measures the proportion of true positives among all predicted positives, and NPV measures the proportion of true negatives among all predicted negatives [26]. These 2 metrics are directly relevant to making clinical decisions about whether specific interventions could be given to patients predicted to be highly suicidal. For an outcome of low prevalence, PPV tends to be low, whereas NPV tends to be high [27]. We then employed a nonparametric approach based on isotonic regression to calibrate the model [28]. The Brier score (equal to 0 under perfect calibration), along with calibration plots, was used to assess model calibration in the test set (i.e., the agreement between observed proportion of positives and mean predicted risk of the outcome in different risk strata) [29]. The top 30 predictors were reported separately for the elastic net penalized logistic regression, random forest, and gradient boosting models. For the neural network, there is no standard solution for ranking predictors. The selection of the top predictors was based on absolute magnitude of predictor coefficient for the elastic net penalized logistic regression and predictor importance score for random forest and gradient boosting models. Predictor importance score measures the contribution of each predictor to the overall prediction and the sum of all scores equals to 100%. Learning curve analysis was performed to evaluate the bias and variance trade-off and to assess if future work would benefit from larger sample size, greater model capacity, or both [30].

Finally, we fitted additional models using predictors restricted to sex, age at the visit, diagnoses and dispensed medications only, and tested for statistical significance ($p < 0.05$, two sided) of decrease in AUC using the method proposed by Hanley and McNeil [31]. These analyses were conducted to explore the predictive potential of the electronic medical records system alone, as it is more feasible to integrate computer algorithms to a single system than create complex linkage between registries for making prediction in real life.

This study is reported as per the transparent reporting of a multivariable prediction model for individual prognosis or diagnosis (TRIPOD) guideline (S1 Checklist). The analyses were planned prior to data retrieval, but we did not register or prepublish the analysis plan. Results outlined in S8–S10 Tables were reported in response to peer review comments. SAS software 9.4 (https://www.sas.com/) and R software 3.6.1 (https://www.r-project.org/) were used for constructing the datasets and descriptive analyses. Scikit-learn (https://scikit-learn.org) and XGBoost (https://xgboost.readthedocs.io/en/latest/python/) packages for Python programming language 3.6.7 (https://www.python.org/) were used for the machine learning analyses during model derivation and evaluation. The code has been placed in a repository on Github (https://github.com/qichense/suicide_ml/).

## Results

Among 541,300 eligible visits to psychiatric specialty care, 18,682 (3.45%) were followed by suicidal outcomes within 90 days and 9,099 (1.68%) within 30 days. Descriptive characteristics of the entire study sample are shown in Table 1.

### Model selection

S5 Table shows the mean (standard deviation) training and validation AUCs of the 4 best models trained via 4 machine learning algorithms, namely elastic net penalized logistic regression, random forest, gradient boosting, and a neural network, as well as the ensemble of different combinations of these best models. S6 Table lists the optimized hyperparameters for each

**Table 1. Baseline characteristics of 541,300 eligible visits by 126,205 patients to psychiatric specialty care during 2011 and 2012.**

| Characteristic | Training set n (%) | Test set n (%) |
|---|---:|---:|
| Visits | 433,024 | 108,276 |
| Inpatient | 49,077 (11.3) | 12,293 (11.4) |
| Outpatient | 383,947 (88.7) | 95,983 (88.6) |
| Unplanned | 94,988 (21.9) | 23,343 (21.6) |
| Planned | 335,494 (77.5) | 84,398 (77.9) |
| Unknown if planned or not | 2,544 (0.6) | 535 (0.5) |
| Unique patients | 100,964 | 25,241 |
| Female | 242,944 (56.1) | 62,355 (57.6) |
| Mean (standard deviation) age at the visit, years | 27.3 (6.1) | 27.2 (6.1) |
| Primary diagnosis[a] | | |
| Substance use disorders | 59,178 (13.7) | 14,427 (13.3) |
| Schizophrenia spectrum disorders | 27,467 (6.3) | 6,073 (5.6) |
| Bipolar disorder | 33,005 (7.6) | 8,412 (7.8) |
| Major depressive disorder | 72,876 (16.8) | 18,676 (17.2) |
| Anxiety disorders | 86,246 (19.9) | 21,933 (20.3) |
| Borderline personality disorder | 17,248 (4.0) | 4,626 (4.3) |
| Attention-deficit/hyperactivity disorder | 53,048 (12.3) | 12,991 (12.0) |
| Autism | 20,831 (4.8) | 5,170 (4.8) |
| Others | 63,125 (14.6) | 15,968 (14.7) |
| Visits followed by | | |
| Suicide attempt/death within 90 days[b] | 14,675 (3.4) | 4,007 (3.7) |
| Intentional self-harm | 13,308 (3.1) | 3,696 (3.4) |
| Self-harm with undetermined intent | 1,277 (0.3) | 353 (0.3) |
| Death from intentional self-harm | 379 (0.1) | 56 (0.1) |
| Death from self-harm of undetermined intent | 164 (0.04) | 39 (0.04) |
| Suicide attempt/death within 30 days[a] | 7,188 (1.7) | 1,911 (1.8) |
| Intentional self-harm | 6,596 (1.5) | 1,775 (1.6) |
| Self-harm with undetermined intent | 505 (0.1) | 142 (0.1) |
| Death from intentional self-harm | 144 (0.03) | 19 (0.02) |
| Death from self-harm of undetermined intent | 55 (0.01) | 14 (0.01) |

[a]Primary diagnosis with an ICD-10 codes ranging from F00 to F99.

[b]Different types of events may occur during the same outcome window.

ICD-10, International Classification of Diagnosis, 10th edition.

model. Results from cross-validation showed that the validation AUCs of the best models of each type were very similar. The ensemble of the 4 best models gave a higher validation AUC and a smaller difference between the training and the validation AUCs relative to the other models and thus was selected and applied to the entire training set to obtain the final model parameters. The subsequent results, therefore, were solely based on ensemble models for both outcomes.

## Model discrimination

The models for predicting suicide attempt/death within 90 and 30 days following a visit to psychiatric specialty care demonstrated good discrimination accuracy. The test AUCs were 0.88 (95% CI: 0.87 to 0.89) and 0.89 (95% CI = 0.88 to 0.90), respectively (Fig 1). At the 95th percentile risk threshold, the sensitivities were 47.2% and 52.9%, the specificities were 96.6% and 95.9%, the PPVs were 34.9% and 18.7%, and the NPVs were 97.9% and 99.1% (Table 2 and S7 Table). Table 2 also shows sensitivity, specificity, and predictive values over a series of risk thresholds.

## Model calibration

Brier scores were estimated to be 0.028 and 0.015 (both close to 0) for the models predicting outcome events within 90 days and 30 days, respectively, indicating good model calibration. The calibration plots (Fig 2) further illustrate high agreement between the observed proportion of positives and mean predicted risk of the outcomes. More details can be found in S9 Table.

## Learning curve analyses: bias versus variance

Fig 3 illustrates the learning curves for assessing the bias and variance trade-off. As the training sample size gradually increases, the 2 curves representing training and validation AUCs end

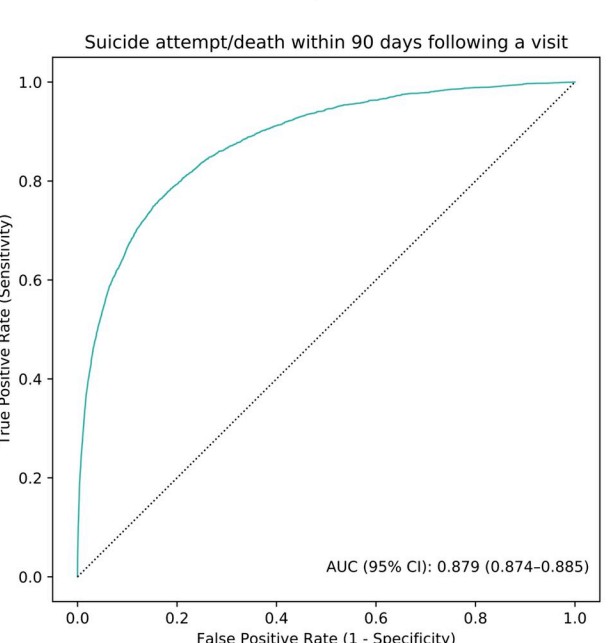
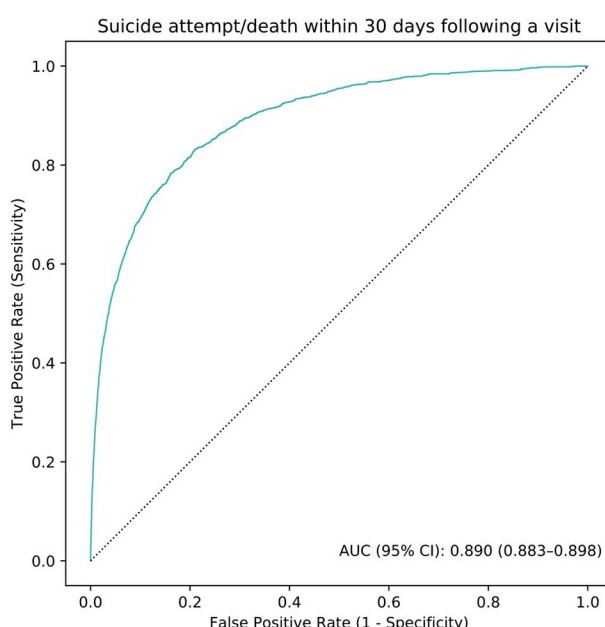

**Fig 1. ROC curves illustrating model discrimination accuracy in the test set for predicting suicide attempt/death within 90 (A) and 30 days (B) following a visit to psychiatric specialty care during 2011 and 2012.** The figure was based on the discrimination accuracy of the ensemble models. The solid line in brown represents the ROC curves achieved by the models. The dotted line in black represents the ROC curves when AUC equals 50%. AUC, area under the receiver operating characteristic curves; ROC, receiver operating characteristic.

**Table 2. Model performance metrics at various risk thresholds for predicting suicide attempt/death within 90 and 30 days following a visit to psychiatric specialty care during 2011 and 2012.**

| Risk threshold | Sensitivity (%) | Specificity (%) | PPV (%) | NPV (%) |
|---|---|---|---|---|
| *Suicide attempt/death within 90 days following a visit* | | | | |
| 99.5th | 10.0 | 99.9 | 74.2 | 96.7 |
| 99th | 17.5 | 99.6 | 64.6 | 96.9 |
| 98th | 27.7 | 99.0 | 51.2 | 97.3 |
| 97th | 36.5 | 98.3 | 45.0 | 97.6 |
| 96th | 42.0 | 97.5 | 38.9 | 97.8 |
| 95th | 47.2 | 96.6 | 34.9 | 97.9 |
| 90th | 62.1 | 92.0 | 23.0 | 98.4 |
| 85th | 71.5 | 87.2 | 17.6 | 98.8 |
| 80th | 77.4 | 82.2 | 14.3 | 99.0 |
| 70th | 85.5 | 72.1 | 10.6 | 99.2 |
| 60th | 90.5 | 61.9 | 8.4 | 99.4 |
| 50th | 94.0 | 51.7 | 7.0 | 99.6 |
| *Suicide attempt/death within 30 days following a visit* | | | | |
| 99.5th | 12.8 | 99.7 | 44.9 | 98.5 |
| 99th | 21.1 | 99.4 | 37.3 | 98.6 |
| 98th | 33.6 | 98.6 | 29.7 | 98.8 |
| 97th | 42.6 | 97.7 | 25.1 | 99.0 |
| 96th | 48.1 | 96.8 | 21.2 | 99.0 |
| 95th | 52.9 | 95.9 | 18.7 | 99.1 |
| 90th | 67.7 | 91.0 | 12.0 | 99.4 |
| 85th | 75.4 | 86.1 | 8.9 | 99.5 |
| 80th | 80.7 | 81.1 | 7.1 | 99.6 |
| 70th | 87.9 | 71.0 | 5.2 | 99.7 |
| 60th | 92.6 | 60.9 | 4.1 | 99.8 |
| 50th | 95.2 | 50.8 | 3.4 | 99.8 |

NPV, negative predictive value; PPV, positive predictive value.

Model performance metrics were based on ensemble models.

up very close to each other and converge at AUCs of 0.88 and 0.89 for the models separately predicting outcome events within 90 and 30 days, suggesting relatively low bias (relatively high AUCs) and low variance (eventual convergence). Because the validation curve is no longer increasing with increased training sample size, future improvements to the model may require more informative predictors and higher model capacity rather than a larger sample size. The convergence of the training and validation AUCs indicates no model overfitting.

## Importance of predictors

Table 3 shows the top 30 predictors with the highest importance based on the best elastic net penalized logistic regression, random forest model, and gradient boosting models, as well as a substantial overlap in top predictors between these models. In general, temporally close predictors tended to be ranked higher than temporally remote predictors. Intentional self-harm during the past 1 year (i.e., within 1 month, 1 to 3 months, 3 to 6 months, or 6 to 12 months), unplanned visit to psychiatric specialty care service during the past 1 to 3 months, diagnosis of borderline personality disorder during the past 3 months (i.e., within 1 month or 1 to 3 months), and diagnosis of depressive disorder during the past month, recent dispensation of

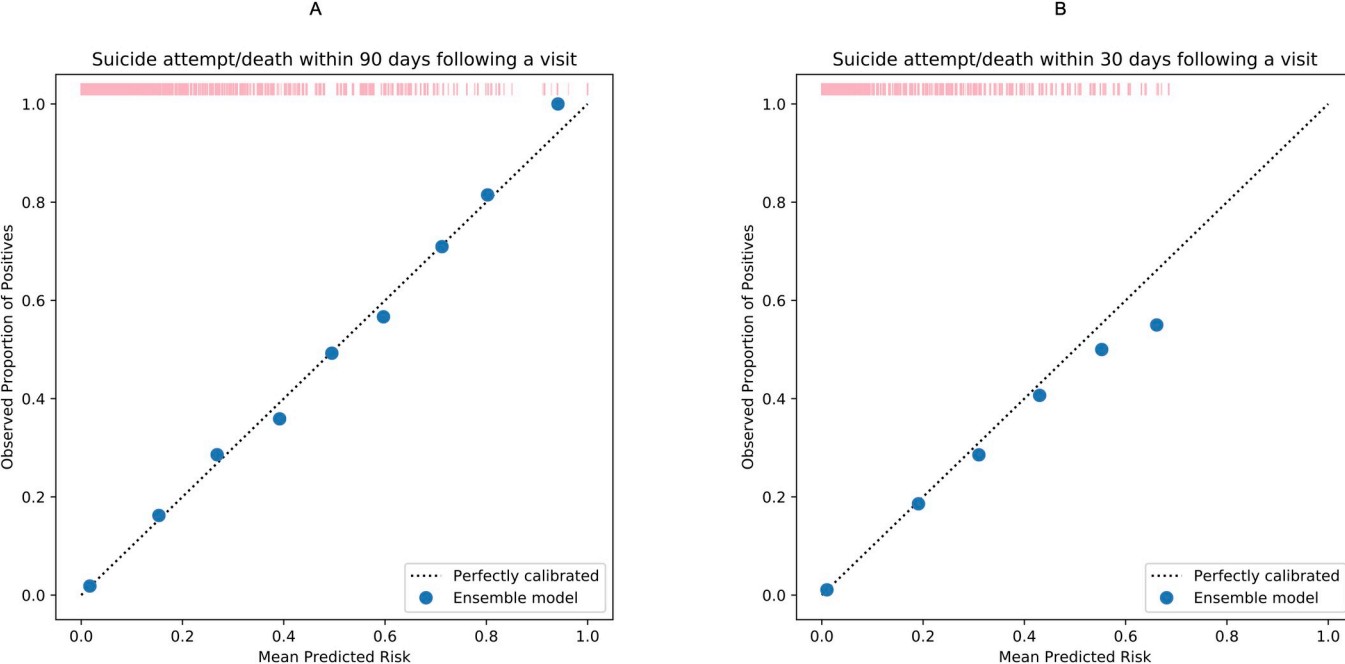

**Fig 2. Calibration plots comparing observed proportion of positives and mean predicted risk suicide attempt/death within 90 (A) and 30 days (B) following a visit to psychiatric specialty care during 2011 and 2012.** The figure was based on the calibration of the ensemble models. Each solid dot in blue represents the observed proportion of index visits followed by a suicidal event in a bin of sample (observed proportion of positives [suicidal events]) against the mean predicted risk in the same bin. More details can be found in S9 Table. The rug plot in pink represents the distribution of the study sample across different predicted risk levels.

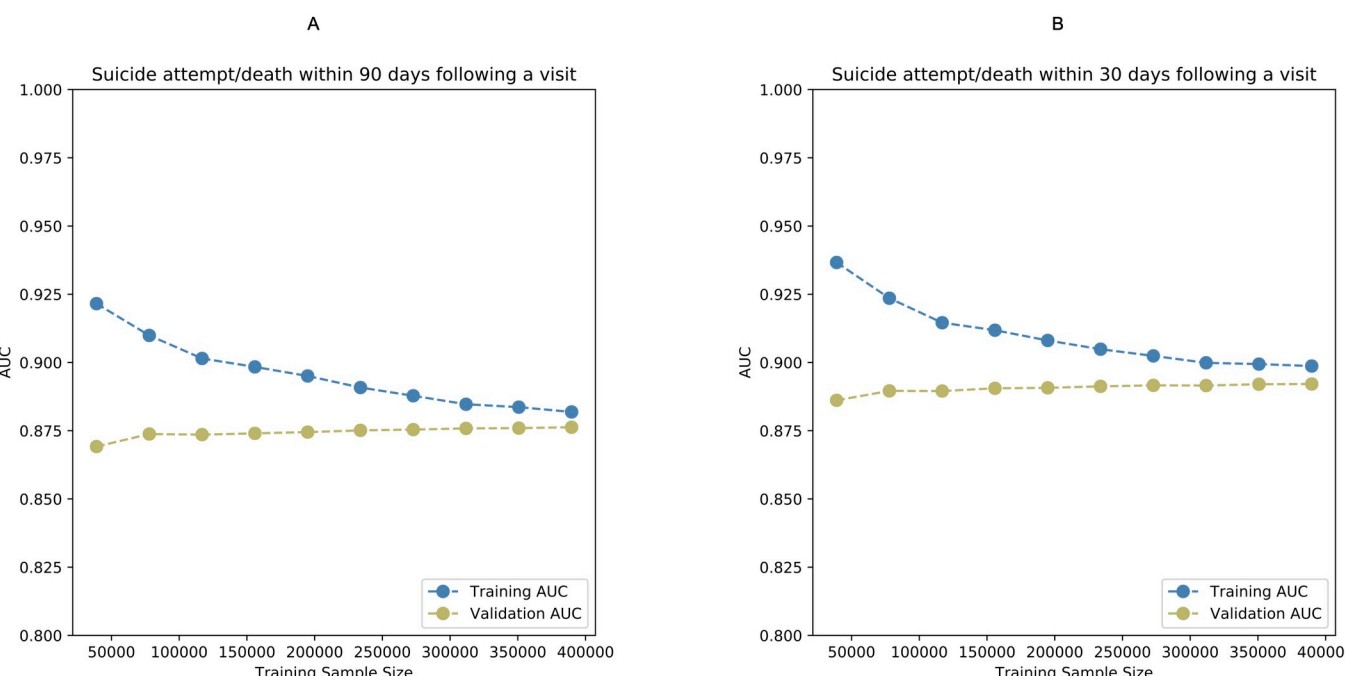

**Fig 3. Learning curves illustrating bias and variance trade-off in the training set for predicting suicide attempt/death within 90 (A) and 30 days (B) following a visit to psychiatric specialty care during 2011 and 2012.** The figure was based on the calibration of the ensemble models. AUC, area under the receiver operating characteristic curve.

**Table 3. Predictors ranked top 30 by the best models of elastic net penalized logistic regression, random forest, and gradient boosting.**

| Predictor | EN90 | RF90 | GB90 | EN30 | RF30 | GB30 |
|---|---|---|---|---|---|---|
| Intentional self-harm 3–6 months before the visit | 1st | 2nd | 1st | 1st | 1st | 1st |
| Intentional self-harm within 1 month before the visit | 2nd | 7th | 6th | 2nd | 5th | 2nd |
| Intentional self-harm 1–3 months before the visit | 3rd | 5th | 3rd | 3rd | 2nd | 3rd |
| Unplanned visit | 4th | 17th | 14th | 8th | 14th | 12th |
| Family history of intentional self-harm | 5th | 1st | 2nd | 6th | 3rd | 5th |
| Intentional self-harm 6–12 months before the visit | 6th | 3rd | 5th | 7th | 4th | 7th |
| Prior intentional self-harm by sharp object | 7th | 8th | 7th | 4th | 9th | 6th |
| Prior intentional self-harm by poisoning | 8th | 4th | 4th | 5th | 6th | 4th |
| Unplanned visit 1–3 months before the visit | 9th | 9th | 10th | 12th | 12th | 11th |
| Unplanned visit within 1 month before the visit | 11th | 6th | 12th | 9th | 7th | 9th |
| Hospitalization within 1 month before the visit | 12th | 12th | 11th | 11th | 8th | 10th |
| Family history of substance use disorder | 13th | 23rd | 17th | 17th | 23rd | 25th |
| Family history of borderline personality disorder | 14th | 16th | 13th | 14th | 15th | 8th |
| Intentional self-harm 1–3 years before the visit | 15th | 10th | 8th | 16th | 10th | 16th |
| Family history of self-harm of undetermined intent | 16th | 24th | 21st | 15th | 24th | 23rd |
| Hospitalization 1–3 months before the visit | 19th | 11th | 9th | 19th | 13th | 14th |
| In-/outpatient visit | 27th | 15th | 16th | 27th | 16th | 20th |
| Family history of anxiety disorders | 26th | 30th | 20th | 22nd | | 30th |
| Planned visit | | 18th | 18th | 10th | 17th | 13th |
| Age at the visit | 10th | | 26th | 13th | | |
| Diagnosis of major depressive disorder within 1 month before the visit | 17th | | 24th | 24th | | |
| Sex | 21st | | 30th | 23rd | | |
| Family history of major depressive disorder | 24th | | 27th | 25th | | |
| Diagnosis of anxiety within 1 month before the visit | 18th | | | 20th | | |
| Intentional self-harm at the visit | 20th | | | 18th | | |
| Diagnosis of ADHD at the visit | 22nd | | | 26th | | |
| Presence of study income | 23rd | | | 29th | | |
| Dispensed benzodiazepines 6–12 months before the visit | 25th | | | | | |
| Diagnosis of other personality disorders than ASPD and BLPD 6–12 months before the visit | 28th | | | | | |
| Diagnosis of substance use disorder within 1 month before the visit | 29th | | | | | |
| Dispensed benzodiazepines within 1 month before the visit | 30th | | | | | |
| Diagnosis of borderline personality disorder within 1 month before the visit | | 13th | 19th | | 11th | 15th |
| Diagnosis of borderline personality disorder 6–12 months before the visit | | 14th | 23rd | | 18th | |
| Diagnosis of borderline personality disorder 1–3 months before the visit | | 20th | 15th | | 22nd | 26th |
| Diagnosis of borderline personality disorder 3–6 months before the visit | | 21st | 22nd | | 19th | 19th |
| Diagnosis of borderline personality disorder at the visit | | 22nd | 29th | | 21st | |
| Diagnosis of borderline personality disorder 1–3 years before the visit | | 25th | 25th | | 27th | |
| Intentional self-harm 3–5 years before the visit | | 19th | | | 20th | |
| Prior self-harm by poisoning of undetermined intent | | 26th | | | 26th | 27th |
| Diagnosis of other borderline personality disorders than ASPD and BLPD at the visit | | 27th | | | | |
| Diagnosis of substance use disorder 3–6 months before the visit | | 28th | | | 30th | |
| Diagnosis of borderline personality disorder 3–5 years before the visit | | 29th | | | 29th | |
| Dispensed antidepressants 3–6 months before the visit | | | 28th | | | |
| Dispensed anxiolytics 6–12 months before the visit | | | | 21st | | |
| Dispensed antipsychotics 1–3 years before the visit | | | | 28th | | |
| Diagnosis of epilepsy 3–6 months before the visit | | | | 29th | | |
| Father's education £ 9 years | | | | 30th | | |

*(Continued)*

**Table 3.** (Continued)

| Predictor | EN$_{90}$ | RF$_{90}$ | GB$_{90}$ | EN$_{30}$ | RF$_{30}$ | GB$_{30}$ |
|---|---|---|---|---|---|---|
| Family history of other borderline personality disorders than ASPD and BLPD | | | | | 25th | |
| Intentional self-harm by unspecified means | | | | | 28th | |
| Diagnosis of substance use disorder 6–12 months before the visit | | | | | | 17th |
| Diagnosis of other personality disorders than ASPD and BLPD 1–3 months before the visit | | | | | | 18th |
| Diagnosis of type 2 diabetes mellitus 1–3 months before the visit | | | | | | 21st |
| Diagnosis of asthma 3–5 years before the visit | | | | | | 22nd |
| Diagnosis of substance use disorder at the visit | | | | | | 24th |
| Diagnosis of epilepsy within 1 month before the visit | | | | | | 28th |

EN: The best elastic net penalized logistic regression model.

RF: The best random forest model.

GB: The best gradient boosting model.

Subscript 90: Model for predicting suicide attempt/death within 90 days following a visit to psychiatric specialty care.

Subscript 30: Model for predicting suicide attempt/death within 30 days following a visit to psychiatric specialty care.

ADHD, attention-deficit/hyperactivity disorder; ASPD, antisocial personality disorder; BLPD, borderline personality disorder.

antidepressants (i.e., 3 to 6 months), anxiolytics (i.e., 6 to 12 months), benzodiazepines (i.e., within 1 month or 6 to 12 months), and antipsychotics (i.e., 1 to 3 years) were ranked as top predictors. In addition, prior intentional self-harm by poisoning or sharp object, family history of suicide attempt, family history of substance use disorder, and family history borderline personality disorder were also ranked as top predictors by more than 1 model. The intercepts and coefficients of the elastic net penalized logistic regression models can be found in S10 Table.

### Predictive potential of electronic medical records system alone

When the candidate predictors were restricted to sex, age at the visit, and those identified from the National Patient Register as well as the Prescribed Drug Register (S1 Table), the AUCs for predicting the outcome events within 90 and 30 days were 0.86 (95% CI: 0.86 to 0.87) and 0.88 (95% CI: 0.87 to 0.88), respectively. Compared with the main models, these decreases in AUCs were statistically significant ($p < 0.001$). S8 Table shows the sensitivity, specificity, and predictive values at different risk thresholds.

## Discussion

To the best of our knowledge, this is the first study using machine learning to determine the potential of the Swedish national registry data for relatively short-term prediction of suicidal behavior in the general psychiatric specialty care. Based on ensemble learning of elastic net penalized logistic regression, random forest, gradient boosting, and a neural network, the final models achieved both good discrimination (AUC was 0.88 [0.87 to 0.89] for the 90-day outcome and 0.89 [0.88–0.90] for the 30-day outcome) and calibration (Brier score was 0.028 for the 90-day outcome and 0.015 for the 30-day outcome).

The AUCs achieved by our models were higher than those of prior studies that predicted suicidal behavior within the 90-/30-day windows in the review of Belsher and colleagues [6]. One model in a prior study among the United States army soldiers with suicidal ideations demonstrated higher AUC (0.93) [32]. The authors of the study, however, did not specify the time window of the predicted outcome. They used cross-validation rather than a separate dataset for the final model evaluation, which tended to overestimate the AUC as indicated by the previous review and simulations [33,34].

To date, there is a lack of agreement on what risk threshold would signal sufficient clinical utility of a prediction model for deployment in clinics. We therefore reported 4 additional model metrics at varying risk thresholds instead of focusing on 1 single preselected threshold. In the current study, at the 95th percentile risk threshold, the model would correctly identify approximately half of all suicide attempts/deaths within 90 days (sensitivity 47.2%); among those psychiatric visits that were predicted to be at high risk, around one-third were actually followed by suicidal events within 90 days (i.e., PPV 34.7%). Although a higher sensitivity could be achieved at a lower threshold, this would be at the cost of a reduced PPV. At the 80th percentile risk threshold, nearly 80% of all suicide attempts/deaths within 90 days would be correctly identified (sensitivity 77.4%); among those visits that were predicted to be at high risk, 1 in 7 was followed by suicidal events within 90 days (PPV 14.3%). Meanwhile, most visits without suicidal events within 90 days would be correctly identified given the high specificity estimates (96.6% and 82.2% at the 95th and 80th percentile risk thresholds, respectively), while most visits predicted to be low risk were not followed by suicidal events given the high NPV (97.9% and 99.0% at the 95th and 80th percentile risk thresholds, respectively). Selecting a reasonable risk threshold for predictive values depends on the resources available for subsequent intervention strategies and their implications for individuals and services. When effective intervention strategies are taken into consideration, model performance metrics at other risk thresholds may also be informative. Future research on cost-effective interventions with negligible or no adverse effects are warranted (e.g., increased frequency of follow-up), as such interventions could be allocated to false-positive patients using models with low PPVs. In addition, prior studies tended to put more emphasis on the role of PPV in assisting clinical decision-making, although NPV might be useful in confirming clinical assessment of low-risk group and helpful in optimizing the allocation of healthcare resource. One prior study, using the Swedish registry data collected during 2001 to 2008 and multivariable regression, developed and validated a prediction model for suicide within 1 year after discharge among in- and out-patients diagnosed with schizophrenia spectrum or bipolar disorder [7]. The final model achieved an AUC of 0.71 on the final validation set. The study prespecified a risk threshold of 1% (i.e., 99th percentile risk threshold), which was close to the prevalence of the predicted outcome (approximately 0.6%), for evaluating the model metrics, although concluded that this risk threshold should not be used clinically, and probability scores were more informative. At the 1% threshold, the sensitivity, specificity, PPV, and NPV were estimated to be 55%, 75%, 2%, and 99%, respectively. Despite a very low PPV, the model achieved a high NPV. Similarly, in the current study, at the risk thresholds close to the prevalence estimates of the predicted outcomes, the NPVs were also high for suicide attempt/death with 90 days (97.8%) and 30 days (98.8%). However, it is difficult to directly compare the models from the 2 studies, given the differences in definition of the predicted outcome (suicide death versus suicide attempt or death) and time window of interest between the studies. Currently, it is unclear under which circumstances a seemingly very high NPV may have clinical utility for an outcome with a low prevalence. Simulation studies are therefore required to determine how NPV varies with the values of PPV, outcome prevalence, and potential cost of intervention. Moreover, screening out on the basis of high NPVs will be limited to the outcome of interest and will not be appropriate if those individuals are at increased risk of other adverse outcomes (e.g., accidents). This suggests that if tools are used in this way, clinicians need to consider risks for other outcomes before these individuals are not further assessed or considered for treatment.

In the current study, the difference in predictive performance between the initial 4 types of models, namely elastic net penalized logistic regression, random forest, gradient boosting, and a neural network, was very small. This is consistent with the findings in the review by Belsher

and colleagues suggesting that no 1 model seems clearly better than another [6]. Future research using the same predictors as in our study may employ elastic net penalized logistic regression only to obtain a relatively better model interpretability at a limited loss of predictive performance. The learning curves suggest that more informative predictors and higher model capacity are likely required to further improve the predictive power. Informative predictors could come from creating transformations of existing predictors (e.g., the frequency of self-harm over a certain time period) or incorporating completely novel predictors such as data from primary medical care services, clinical documentation text, audio and video data on clinical interview, vital physiological parameters continuously monitored via wearable devices. Novel and previously untested data may open opportunities for deep learning analytics to improve prediction of suicidal behavior through identifying highly complex data patterns [35,36]. Deep learning analytics may also improve model capacity by creating a better representation of the predictors used in the current study [37].

There was a substantial overlap in top-ranked predictors between different models. While borderline personality disorder, substance use disorder, depression, dispensed benzodiazepines and/or antidepressants have already been identified as risk factors for suicide [38,39], the timing of these factors may play a vital role in predicting subsequent suicidal behavior [40]. Our study showed that temporally close predictors tended to be ranked higher than distal ones. These results call for more research in the timing of diagnoses of psychiatric disorders and the use of psychotropic medications in relation to suicidal behavior. Using the Danish registry data, a recent study predicting sex-specific suicide risk reported that diagnoses occurring long (e.g., 48 months) before suicide were more important for suicide prediction than diagnoses occurring shortly (e.g., 6 months) before suicide [41]. Since the authors defined the predictors using time of suicide, which could not be known beforehand, the models would not be implementable in clinical systems. One Swedish registry-based cohort study of 48,649 individuals admitted to hospital after suicide attempt found that prior hanging, drowning, firearms or explosives, jumping from a height, or gassing at suicide attempt better predicted subsequent suicide than poisoning and cutting [42]. Our study, however, showed that poisoning and cutting were ranked higher than other methods when predicting a broadly defined behavior including both suicide attempt and suicide. As expected [43], family history of several psychiatric disorders (i.e., anxiety disorders, borderline personality disorder, major depressive disorder, substance use disorder) were also among the top predictors, whereas family history of somatic disorders were not. It should be noted, however, that in predictive modeling, the importance of a predictor reflects the extent to which permuting the value of the predictor will increase prediction error [44]. Unlike causal risk factors, predictors with higher importance may or may not have a direct impact on the outcome. Highly correlated predictors, differing in model-based importance scores, may have very similar univariate predictive ability but the one that is less predictive would be given low importance since its importance value is conditional on the highly correlated predictor being in the model. The coefficients in the elastic net penalized logistic regression, unlike in ordinary logistic regression, were shrunk in magnitude. Although the size of coefficients could be used for ranking predictor importance, it does not represent the effect size of a specific predictor on suicidal behavior. Moreover, the sign (positive/negative) of coefficients should not be interpreted as an increase or decrease in risk of suicidal behavior. This is because, in predictive modeling, the predictors are not necessarily causal risk factors of the predicted outcome and the direction of effect of a specific predictor is influenced by other model selected predictors that are not necessarily confounders. Of more relevance in predictive modeling is the overall predictive performance rather than individual model coefficients. Accurate estimation of the magnitude and direction of risk factors requires different research designs, such as cohort or cross-control investigations. When candidate

predictors were restricted to those related to patient's history of diseases and dispensed medications only, the AUCs were only slightly worse than those achieved by the models using all predictors obtained via complex linkages among registers, suggesting limited incremental value of the predictors from other registers than the National Patient and the Prescribed Drug Registers in predicting suicidal behavior. The results have pragmatic implications, because it is more feasible to integrate data from within the electronic medical records system than it is to create complex linkages among registries.

Our study is subject to several limitations. To ensure reasonable quality of data on both individual health and social economic information, as well as family history of disease and crime, the sample was limited to patients who were adults at the visit and covered by the Swedish Medical Birth Register. By the end of year 2012, the oldest in the study population were 39 years old. The derived models therefore are less likely to generalize well to adult patients older than 40 years or to children and adolescents. The candidate predictors were limited by the data sources. The derived models are not clinically based as hospital services are not in a position to link disparate registers. Our outcomes have limited validation. Nevertheless, according to an external review on the validity of diagnoses in the Swedish national inpatient register [13], the PPVs of the register-based diagnostic codes varied between 85% and 95%. Notably, the PPV of injury, including both accidental injury and self-injury, was 95%. On the basis of this, we think that the PPV of our outcome is likely to be high, despite a lack of specific validation study. Our definition of suicidal behavior was based on the ICD-10 criteria, which does not make distinction between suicidal and nonsuicidal self-harm. Although this definition has been widely used in prior research, whether nonsuicidal self-harm represents actual suicide risk remains controversial. A certain amount of nonsuicidal self-harm events were labeled as positive outcome events, which may have led to an overestimation of the PPVs. On the other hand, underestimation of the PPVs is also possible as planned visits with a recorded self-harm were not labeled as positive outcome events. When defining the outcome, we assumed that unplanned inpatient or outpatient visits after the index hospitalization with a recorded self-harm were incident suicidal events. This was based on the selection of the study sample, which was restricted to patients who, at the index visit, received a primary diagnosis with an ICD-10 code ranging from F00 to F99. This means that the index visits were not primarily due to suicide attempt/self-harm (which has a different ICD code, i.e., ICD-10: X60–X84 and Y10-Y34), and thus, the identified suicidal events after the index visit were very unlikely to be the same visit as the index visit. Future validation studies could examine how accurately identified events in the Patient Register represent true incident suicidal events. The preliminary selection of 425 candidate predictors and their time windows was somewhat arbitrary. In theory, many more candidate predictors can be generated from the registry data or via transformation of existing ones. As a result, it is unclear whether the achieved model performance in the current study is the best achievable performance. Our models have not yet been externally validated, and thus, the generalizability of the models to other populations remains unknown. In future work, guidelines will be critical in improving transparency and reproducibility of research in predictive modeling and should be followed once they are finalized.

In our study, the models were derived to predict suicidal behaviors rather than the rare outcome of suicide deaths. In relation to deaths, the model performance would be quite different with higher specificities and lower sensitivities, and the calibration would be poor. As some metrics of discrimination will be low, 1 promising approach would be to solely use probability scores rather than risk thresholds to investigate suicide deaths [7]. Future research could compare ordinary logistic regression models with machine learning models, and in particular the former can focus on the top-ranked predictors identified in this study.

## Conclusions

By combining the ensemble method of multiple machine learning algorithms and high-quality data solely from the Swedish registers, we developed prognostic models to predict short-term suicide attempt/death with good discrimination and calibration. Whether novel predictors can improve predictive performance requires further investigation.

## Supporting information

**S1 TRIPOD Checklist. Prediction model development.**
(PDF)

**S1 Fig. Flowchart for study sample identification.**
(TIF)

**S1 Table. A total of 425 candidate predictors generated from the Swedish national registers.**
(DOCX)

**S2 Table. International Classification of Diseases (ICD) codes for identifying clinical diagnoses from the National Patient Register.**
(DOCX)

**S3 Table. International Classification of Diseases (ICD)-10 codes for identifying methods used for prior self-harm from the National Patient Register.**
(DOCX)

**S4 Table. Prior use of medications identified from the Prescribed Drug Register according to the Anatomical Therapeutic Chemical (ATC) Classification System.**
(DOCX)

**S5 Table. Training and validation AUCs obtained from 10-fold cross-validation for the best models trained by each selected machine learning algorithm and the ensemble of different combinations of these best models.**
(DOCX)

**S6 Table. Optimization of hyperparameters using grid search.**
(DOCX)

**S7 Table. Number of true vs.** predicted outcome at the 95th percentile risk threshold.
(DOCX)

**S8 Table. Model performance metrics at various risk thresholds for predicting suicide attempt/death within 90 and 30 days following a visit to psychiatric specialty care during 2011 and 2012, predictors being restricted to sex, age at the visit, and those identified from the National Patient Register as well as the Prescribed Drug Register.**
(DOCX)

**S9 Table. Statistics underlying model calibration curves for predicting suicide attempt/death within 90 and 30 days following a visit to psychiatric specialty care during 2011 and 2012.**
(DOCX)

**S10 Table. Elastic net logistic regression model selected predictors and coefficients.**
(DOCX)

## Author Contributions

**Conceptualization:** Qi Chen.

**Formal analysis:** Qi Chen.

**Funding acquisition:** Seena Fazel.

**Investigation:** Qi Chen.

**Methodology:** Qi Chen, Yanli Zhang-James, Eric J. Barnett, Stephen V. Faraone, Seena Fazel.

**Project administration:** Qi Chen.

**Resources:** Henrik Larsson.

**Supervision:** Paul Lichtenstein, Stephen V. Faraone, Henrik Larsson, Seena Fazel.

**Visualization:** Qi Chen.

**Writing – original draft:** Qi Chen.

**Writing – review & editing:** Qi Chen, Yanli Zhang-James, Eric J. Barnett, Paul Lichtenstein, Jussi Jokinen, Brian M. D'Onofrio, Stephen V. Faraone, Henrik Larsson, Seena Fazel.

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
