## [Editor Report · Decision Letter 0]

15 Jan 2020

Dear Dr Chen, 

Thank you for submitting your manuscript entitled "Predicting suicide attempt/death following a visit to psychiatric specialty care by applying machine learning to the Swedish national registry data" for consideration by PLOS Medicine.

Your manuscript has now been evaluated by the PLOS Medicine editorial staff and I am writing to let you know that we would like to send your submission out for external peer review.

**Please be aware that, due to the voluntary nature of our reviewers and academic editors, manuscript assessment may be subject to delays during the holiday season. Thank you for your patience.**

Kind regards,

Helen Howard, for Clare Stone PhD

Acting Editor-in-Chief

PLOS Medicine

plosmedicine.org

---

## [Decision Letter · Decision Letter 1]

11 Feb 2020

Dear Dr. Chen,

Thank you very much for submitting your manuscript "Predicting suicide attempt/death following a visit to psychiatric specialty care by applying machine learning to the Swedish national registry data" (PMEDICINE-D-20-00084R1) for consideration at PLOS Medicine. 

[LINK]

In light of these reviews, I am afraid that we will not be able to accept the manuscript for publication in the journal in its current form, but we would like to consider a revised version that addresses the reviewers' and editors' comments. Obviously we cannot make any decision about publication until we have seen the revised manuscript and your response, and we plan to seek re-review by one or more of the reviewers. 

We expect to receive your revised manuscript by Mar 03 2020 11:59PM. Please email us (plosmedicine@plos.org) if you have any questions or concerns.

We look forward to receiving your revised manuscript. 

Sincerely,

Caitlin Moyer, Ph.D.

Associate Editor 

PLOS Medicine

plosmedicine.org

Specific point – You will note that is repeatedly raised among referees that the study is not reproducible and that without the code and parameters for others to replicate this study is severely limited. We are unable to consider a resubmission unless the code is placed in a repository such as Github and all parameters and associated data made freely available. This is regardless of other revisions undertaken. 

General points: 

Abstract – please add some demographic information to the abstract as well as mean ages of male and female; please add p values where 95% Cis are given; please provide a sentence on the limitations of the study as the final sentence of the ‘Methods and Findings’ section; ‘models outperforming existing models in predicting relatively short-term suicide attempt/death.’ (also page 10) – unless you perform side-by-side comparisons and show superiority you need to remove such statements. Any comparisons added in revision would be assessed by referees. 

Data – I would remove this ‘they are subject to secrecy I’ and on resubmission state where code can be accessed and this needs to be live for reviewers to access. 

Page 7 – please remove the URL l (http://vassarstats.net/roc_comp.html) and simply say the name of the test and provide the reference. 

Comments from the reviewers:

Reviewer #1: This is a well-written manuscript, analyzing big data on suicide attempt/death to develop prediction models based on several machine learning algorithms and describing its methodology in detail. Particularly, models developed in this study showed relatively higher PPV despite very low prevalence of suicide outcome, compared with other machine learning studies. My comments are as follow.

1. In this study, suicide attempt was defined as intentional self-harm (ICD-10: X60-X84) or self-harm of undetermined intent (ICD-10: Y10-Y34). Considering that the study population were young adults, a considerable portion of self-harm events might correspond to non-suicidal self-injury. It is controversial whether non-suicidal self-injury represents actual suicide risk. Although the authors already mentioned self-harm behavior in the discussion section as limitation, non-suicidal self-injury should be addressed to clarify suicide outcome more clearly.

2. Prediction models based on machine learning algorithms may be useful in screening at risk population. But, models developed in this study had archived relatively low sensitivity despite high specificity and good AUC performance.

Reviewer #2: This paper reports the development of algorithms to predict suicide-related outcomes using data from a large registry. The approaches taken appear sound to me, and the contribution to knowledge is important, given the relatively high predictive performance achieved compared to other initiatives. The authors are appropriately aware of the limitations here, in particular the need for cross-site algorithm replication and the distance between the data resources used here and real-world health record capabilities. My comments are minor. 

1. The section in the Results on predictive potential of electronic medical records alone felt a little under-described (e.g. it would have been helpful to have the standard list of performance metrics rather than just the AUCs), and the final sentence on p9 looks unfinished. This is an important section because it is likely to represent the element that might most readily be replicated elsewhere, so it would be helpful to have more detail. 

2. I think it might be helpful if the text in the 3rd paragraph of the discussion was re-ordered so that the sensitivity and PPVs (and possibly NPVs) of the 95th and 80th percentile alternatives were more explicitly linked and discussed together, as these are most reflective of potential clinical applicability. 

3. On p11, the statement about NPV being more useful for directing resources is a little problematic, as the consideration here is just focused on a single, albeit important, outcome. In order for NPV to be used in this way, it would have to be established that patients falling into the negative group were at lower risk of all other adverse outcomes; otherwise they might receive suboptimal care just on the grounds of lower suicidal risk. 

4. Although I can understand the reason for restrictions on data access, it would be helpful to have a more explicit statement from the authors about what they intend to do with their algorithms. Clearly there is a need for replication here, but I don't believe that this is going to be possible from the information provided. 

Reviewer #4: "Predicting suicide attempt/death following a visit to psychiatric specialty care by applying machine learning to the Swedish national registry data" trained and validated various machine learning models, on a comprehensive Swedish dataset of 541,300 visits from 126,205 patients. The final ensemble model achieved AUCs of about 0.88-0.89, on predicting 30-day and 90-day suicide attempts. Various ancillary analyses were made to establish the degree of calibration, impact of additional training data and of the electronic medical records predictor subset.

In particular, the claimed PPVs from the validated model are very promising (0.349/0.187, with corresponding sensitivity of 0.472/0.529, at the 95th percentile risk threshold for 90-day/30-day outcomes respectively); a recent review (Belsher et al, JAMA Psychiatry, citation 6 in the manuscript) reported that most previous prediction models had extremely low PPVs of <0.01. That said, it should be noted that expected PPV appears to be significantly affected by the population the model(s) are applied to, with general populations with very low attempts/mortality being particularly challenging. As such, the performance achieved in this manuscript might be considered in the context of the target population already seeking psychiatric specialty care. Nonetheless, it remains an interesting study from the scale involved, and the comprehensiveness of the data features available, made possible by the quality of the Swedish national registers. Separation between training and test data also seems well-respected.

There however remain a number of issues that might be addressed:

1. While the study focuses on various machine learning models for data analysis (in particular: elastic net penalized logistic regression, random forest, gradient boosting [XGBoost], neural network), there are next to no pertinent details provided about these methods. For example, model hyperparameters are acknowledged as being grid-searched over, but there is no mention about what these hyperparameters were, nor their ranges (e.g. for random forests, how many trees, what were the tree max depths, etc? for neural networks, what was the architecture, activation functions, learning rate used, etc?) Although the datasets cannot be made public, there appears no reason why a reproducible description of the machine learning models cannot be described in the supplementary material.

2. The detailed training and validation AUC results presented in eTable 5 suggests that the relatively high PPV/AUC performance is largely due to the specific data examined, and less the specific state-of-the-art machine learning models used. In particular, logistic regression alone (albeit elastic net penalized, EN) achieved validation AUCs of 0.8721/0.8883 for 90-day/30-day suicide attempt prediction. The improvement with an ensemble of all four models (to 0.8751/0.8909 respectively) is extremely small. As such, it appears reasonable for the authors to focus on EN here, given it maintains the interpretability of standard logistic regression (e.g. via odds ratios; refer for example "Machine learning algorithms for outcome prediction in (chemo)radiotherapy: An empirical comparison of classifiers", Deist et al., Med Phys 2018), as opposed to the other methods.

3. It appears that the self-harm predictors were binary ("Age at the visit and family income were treated as continuous variables and rescaled to the range of 0-1. The other predictors were treated as binary or categorical variables"); was information about the frequency of self-harm episodes within each time period available, and if so, might this frequency be reasonably considered?

4. Given the greater importance of PPV due to the large number of negatives, the authors might consider exploring precision-recall curves and reporting the AUPRC metric too (e.g. alongside the ROC curves in Figure 1).

5. For Table 1, the figures in brackets might be explained.

6. For eFigure 1, the sum of first-stage exclusion (1028907+210250) and qualifying visits after that exclusion (570945) does not appear to tally with the original total (1810602).

7. For eTable 1, the number of predictors accounted for at finer levels of granularity might also be listed, for convenience (e.g. "Diagnosis at the visit [37 predictors?]")

8. eTable 6 further suggests that the bulk of predictive performance might be attained from a relatively small subset of predictors. In particular, "Intentional self-harm 3-6 months before the visit" was the top predictor for nearly all models, with "Family history of intentional self-harm" and "Intentional self-harm 1-3 months before the visit" probably the next most consistently important predictors. Given this, the authors might examine whether predictors relating to individual/family history of self-harm alone would yield comparable performance to the full 425-predictor models, or indeed the electric medical record-only predictor subsets (which again are able to achieve similar AUCs of 0.86/0.88).

9. The authors might consider providing another table with the data from eTable 6, but sorted by (mean?) rank to show predictors in general order of importance.

10. In the Model derivation and evaluation section, it is stated that "Based on the results of cross-validation, the model giving the highest average validation AUC and showing the smallest difference between the training and validation AUCs was selected and applied to the entire training set to obtain the final model parameters". However, the model having the highest average validation AUC might not necessarily also show the small difference between training and validation AUCs. How were these objectives balanced?

11. A "risk threshold" at various percentiles is referenced frequently, and appears to be achievable by varying the operating point of each machine learning model. There might be a brief explanation of what the various percentiles of risk threshold signify.

12. Certain assertions appear too definite, e.g. "Because the validation curve is no longer increasing with increased training sample size, we can conclude that future improvements to the model will require more informative predictors and higher model capacity rather than a larger sample size".

13. Some analysis/comment on prediction of attempts vs. prediction of deaths would be interesting.

14. Finally, there are some minor textual issues:

"The Medical Birth Register covers nearly all deliverys in Sweden..." -> "nearly all deliveries"

"Despite statistically significant decrease in the value (p<0.05)." -> incomplete sentence fragment?

In summary, while the authors have systematically investigated various ensemble combinations of machine learning models on a large set of 425 predictors in their main analyses, the presented evidence suggests that virtually the same performance might have been achieved with just logistic regression and a much smaller set of the most informative predictors relating to individual/family history of self-harm. Therefore, while the contribution to suicide prediction remains welcome, the applicability of less-interpretable machine learning models (random forests, XGBoost, neural networks) seems not yet well established, for this particular study.

Reviewer #5: The authors applied an ensemble learning method including four types of models to predict suicide attempt/death within a 90-days/30-days window for patients aged 18-39 after a visit to psychiatric specialty care using Swedish national registry data and electronic health records (EHR). 425 candidate predictors were carefully selected covering all relevant information in the registry. The models were trained using a large sample size of patient visits and model parameters were selected using cross-validation. Results showed that the ensemble model outperformed existing models on discrimination and calibration, with high AUC, specificity and negative predictive values. Main findings include a comparison of top 30 predictors from each model, and prediction result using only EHR data also had good performance. 

Strength: The study investigated an urgent and important topic in mental health. The authors applied a suite of machine learning methods to high quality national registry data. Analyses were well done with clear communication of their findings. The top predictors could provide insightful information to practitioners. 

Weakness: The developed ensemble model has not been externally validated. So it is unclear how well it will perform prospectively or in a different patient population. Sensitivity and positive predictive values are relatively low, which decrease the potential impact of this model as a prevention/monitoring tool. 

I have the following comments to the authors:

1. It is unclear why these four particular types of machine learning models were selected in this study. Could you provide a brief rationale and explanation of the advantage/disadvantage of each type of model? Since they performed similarly, why include all four models?

2. It is unclear how missing data on the predictors were handled in the study. Could you provide the procedure for handling missing data? Were there any imputation done?

3. Why did you select 80th-99th risk thresholds in Table 2?

4. The predictors listed in eTable 6 are very important results and should not be in the appendix. How about swapping with Table 3 or Figure 3 which is not very interesting?

5. The paper is written for an audience with background in machine learning and know the concepts of model discrimination and calibration. It would be helpful to include the definition of these concepts when they first appear for a broader readership. For example, the explanations on sensitivity and specificity in 3rd paragraph on page 10 should appear earlier in the paper.

6. The last sentence on page 9, starting with "Despite statistically…", is unclear what it is referring to.

[LINK]

---

## [Decision Letter · Decision Letter 2]

21 May 2020

Dear Dr. Chen,

Thank you very much for submitting your revised manuscript "Predicting suicide attempt/death following a visit to psychiatric specialty care by applying machine learning to the Swedish national registry data" (PMEDICINE-D-20-00084R2) for consideration at PLOS Medicine. 

Your paper was re-evaluated by a senior editor and discussed among all the editors here. It was also sent to three of the original reviewers, including a statistical reviewer. The reviews are appended at the bottom of this email and any accompanying reviewer attachments can be seen via the link below:

[LINK]

In light of the comments of Reviewer 3, I am afraid that we will not be able to accept the manuscript for publication in the journal in its current form, but we would like to consider a further revised version that addresses the reviewers' and editors' comments. Obviously we cannot make any decision about publication until we have seen the revised manuscript and your response, and we plan to seek re-review by one or more reviewers. 

We expect to receive your revised manuscript by May 28 2020 11:59PM. Please email us (plosmedicine@plos.org) if you have any questions or concerns.

We look forward to receiving your revised manuscript. 

Sincerely,

Caitlin Moyer, Ph.D.

Associate Editor 

PLOS Medicine

plosmedicine.org

1.Response to reviewers: Please respond to the remaining concerns of Reviewer 3. Please describe in the text the details for the model parameters and equations (these can be supplemental files) and if possible/relevant please include a diagram illustrating the models.

2.Data Analysis Plan: Did your study have a prospective protocol or analysis plan? Please state this (either way) early in the Methods section.

3.Abstract: Please make the last sentence of the Methods and Findings section (on limitations) more transparent; based on what you wrote in the Discussion we suggest: “A limitation of our study is that our models have not yet been externally validated and thus the generalizability of the models to other populations remains unknown.” or similar.

4. In the first paragraph of the Discussion, AUC values with subscripts “30” and “90” are used, presumably to represent the AUC corresponding to 30 days and 90 days post-visit predictions, however these abbreviations are not defined earlier in the text.

5. Table 1: Please define the abbreviation for “SD” in the legend.

6. Table 3: Please make it clear in the legend that the “30” and “90” subscripts represent the models at 30 and 90 days post-visit, respectively.

7. Figure 1 and Figure 2: In the figure legends, please describe what each line in the plot represents.

8. Checklist: Please ensure that the study is reported according to the TRIPOD guideline, and include the completed TRIPOD checklist as Supporting Information. When completing the checklist, please use section and paragraph numbers, rather than page numbers. Please add the following statement, or similar, to the Methods: "This study is reported as per the transparent reporting of a multivariable prediction model for individual prognosis or diagnosis (TRIPOD) guideline (S1 Checklist).”

Comments from the reviewers:

Reviewer #3: Thank you to the authors for your careful response to reviewer comments and clarification to many questions. I still have some outstanding concerns which I detail below. My biggest concern is that of items 1 and 2. While the code is helpful for other who want to do the same analysis, it is not helpful for assessing the performance of these models in other settings, which should be the goal of this work. Additionally, without more described quality test on the outcome, I have concerns that some self-harm events are actually events from the index visit that due to administrative reasons in the data base appear as another visits, thus falsely giving the impression of high performance, especially for unplanned index visits. 

1. While the authors have made the code available on github, which is great, the models themselves are still not available. At a bare minimum the authors should make the logistic regression model available. All variables selected and the coefficients that goes with those variables. The code itself has limited use without the data it was run on, which for good reasons is unavailable to readers. 

2. I am still concerned about measurement error following an unplanned visit. The authors should report the proportion of unplanned visits in Table 1. Have authors done data quality checks on the self-harm outcome after unplanned visits? For example, it is common for ED visits that lead to a hospitization to be counted as two separate visits, these would both be unplanned. Would the code associated with a hospitalization in this case be counted as an outcome for the ED visit? Additionally, ED visits can cover more than one day, if this is the case does the data you have on hand ensure that ED visits that stretch over more than one day are coded as one visit?

a. Small comment for clarification: I would recommend changing this sentence: "Planned visits were likely to be follow-up healthcare appointments following an incident suicide attempt and thus were not included."

To

Planned visits were likely to be follow-up healthcare appointments following a suicide

attempt and thus were not classified as suicide attempts for our analysis.

If I understand correctly unplanned visits were included in the analysis as non-self-harm events, by saying they were not included it sounds like they were removed from the analysis altogether. 

3. I remain concerned about differences in performance of the model in predicting suicide attempt after unplanned and planned visits. In my previous comments I referred to unplanned visits as inpatient/ED visits, I much prefer the terminology "unplanned" - thank you! I remain concerned that many outcomes after an unplanned visit are actually the same visit or care for the same self-harm event. Without some information about the QA performed to ensure that self-harm events after an unplanned visit are true self harm events (rather than and ED visit lasting two days and billed as such or and ED visit followed by a hospital visit etc.) it is difficult to assess if this performance is true (I hope it is this would be great to get into clinical practice if it proves replicable!) or due to misclassification of the outcome. In the list of predictors, a top predictor across all models is "unplanned" visit - which is another reason I am concerned about this. 

4. The calibration plots presented in the paper need to be described more carefully, the authors respond to my question about the plots in the response to reviewer comments #12 on page 10, but this is not described in the paper and the general audience for this paper is not computer scientists who are the people the authors say are most used to looking at these plots. The plots need to be described much more carefully as to be as a statistician this is not describing calibration. To me, and in the statistical literation, calibration is how well your predicted probability aligns with observed probability. I think this is more informative for assessing performance as well. It is straight forward to look at the average predicted probability in a bin compared to the average observed probability. This is what I would expect in a calibration plot. The current calibration plot I do not find helpful, especially since the definition of a "positive" is not well-described.

5. Table 1 is really helpful to look at. While self-harm like represents a small proportion of diagnosis for the index visit, it is very relevant in this context. I think it would be good to add the proportion of visits with a self-harm primary diagnosis to Table 1. 

6. I appreciate the authors efforts to add understanding of the models, by including a table with top predictors of the models. The authors should cite which variable important measure was used for ranking predictors in the random forest and gradient boosting models. I am assuming the magnitude of coefficient size was use for logistic regression, but it would be useful to provide this information. Are predictors listed in Table 3 only predictors of increase risk or are some predictors of decreased risk as well? It might be worth elaborating on that in the table or text. 

Reviewer #4: The authors have adequately responded to our previous comments. The availability of the code on GitHub is appreciated. The major reservation remains that standard (elastic net penalized) logistic regression produces performance comparable to the other, more-complex and less readily interpretable machine learning techniques. We do however agree with the authors that the above observation on the efficacy of logistic regression could not have been known without empirical evaluation.

Reviewer #5: The authors did a good job addressing my comments and concerns. I am satisfied with the revision and have no additional comments.

[LINK]

---

## [Decision Letter · Decision Letter 3]

28 Jul 2020

Dear Dr. Chen,

Thank you very much for submitting your revised manuscript "Predicting suicide attempt/death following a visit to psychiatric specialty care by applying machine learning to the Swedish national registry data" (PMEDICINE-D-20-00084R3) for consideration at PLOS Medicine. 

Your paper was evaluated by a senior editor and discussed among all the editors here. It was also seen again by one of the reviewers, and the comments are appended at the bottom of this email. In the report, Reviewer 3 notes that you have addressed most of the issues raised; however, additional clarification is requested pertaining to the construction of the calibration plots.

Although we will not be able to accept the manuscript for publication in the journal in its current form, we would like to consider a revised version that addresses the remaining points raised by Reviewer 3 and the editors' comments. Obviously we cannot make any decision about publication until we have seen the revised manuscript and your response. 

We expect to receive your revised manuscript by Aug 04 2020 11:59PM. Please email us (plosmedicine@plos.org) if you have any questions or concerns.

We look forward to receiving your revised manuscript. 

Sincerely,

Caitlin Moyer, Ph.D.

Associate Editor 

PLOS Medicine

plosmedicine.org

1.Response to reviewers: Please fully respond to the comments of reviewer 3, including whether the calibration plot was derived from the training data set or the validation set and in the methods or the legend of Figure 2 please describe how the plots were created/what is represented in the plots.

2.Please revise your title according to PLOS Medicine's style. Your title must be nondeclarative and not a question. It should begin with main concept if possible. Please place the study design ("A randomized controlled trial," "A retrospective study," "A modelling study," etc.) in the subtitle (ie, after a colon). We suggest: “Predicting suicide attempt or death following a visit to psychiatric specialty care: A machine learning study of Swedish national registry data” or similar.

3.Abstract: Methods and Findings: If possible please present the p values for both the 30 day and 90 day models

4.Author Summary: First bullet point under “What do these findings mean?”: Please revise to: Our findings suggest that combining machine learning with registry data has potential to accurately predict short -term suicidal behavior.

5.Discussion: Middle of paragraph on page 13: Please rephrase the term “completed suicide” in the following sentence; we suggest: “However, it is difficult to directly compare the models from the two studies, given the differences in definition of the predicted outcome (suicide death vs suicidal attempt or death) and time window of interest between the studies.” or similar.

6.Figure 1: Please change the colors/patterns of the solid and dotted lines to make them easier to differentiate.

7.Supporting information file: eTable 7: The word “days” is missing from the legend following “90” and “30”

8.TRIPOD Guideline: S1 Checklist is not present in the file inventory, please provide the TRIPOD checklist. When completing the checklist, please use section names and paragraph numbers to refer to locations within the text, rather than page numbers.

Comments from the reviewers:

Reviewer #3: The authors have done a good job in making clarifications and addressing reviewer concerns. The additional limitations added to the discussion are important. 

I am still confused by the calibration plot. I cannot find in the paper if the calibration plot is created using the training data or the validation data set. It should be created in the validation data set using percentile bins from the training data. It appears that deciles were used for the calibration plot (but why are there are only 9 dots instead of 10 ?). I am still very surprised that the observed probability of a suicide attempt in the highest risk group is 100%. The math doesn't make sense here, because if this was created in the validation data set, then there would be about 10,000 visits in the highest risk decile; given the graph it says that nearly 100% of those visits were observed to have a suicide attempt following the visits. that would be about 10,000 suicide attempts, but there should only be about 3,726 suicide attempts in the entire validation data set. The math continues to be a problem with if the calibration plot was created with the training data set.

Please provide more details (not just the function that was used) on how these calibration plots were created.

A common approach is to divide your visits into deciles, these deciles are on the x axis with the mean predicted risk in that percentile. Then on the y-axis is the observed proportion of visits followed by a suicide attempt. At the end of the day a calibration plot needs to indicate in specific bins of people defined by risk, how similar is their predicted risk (from the model) and their observed risk (proportion of those visits with an event following the visit).

[LINK]

---

## [Decision Letter · Decision Letter 4]

17 Sep 2020

Dear Dr. Chen,

Thank you very much for re-submitting your manuscript "Predicting suicide attempt or death following a visit to psychiatric specialty care: A machine learning study of the Swedish national registry data" (PMEDICINE-D-20-00084R4) for review by PLOS Medicine.

I have discussed the paper with my colleagues and the academic editor and it was also seen again by one of the reviewers. I am pleased to say that provided the remaining editorial and production issues are dealt with we are planning to accept the paper for publication in the journal.

[LINK]

We look forward to receiving the revised manuscript by Sep 24 2020 11:59PM. 

Sincerely,

Caitlin Moyer, Ph.D.

Associate Editor 

PLOS Medicine

plosmedicine.org

Requests from Editors:

1.Reviewer comments: Supporting information file eTable 9: As suggested by reviewer 3, please do update the table with the number of unique individuals at each row.

2. Title: Please revise the title: “Predicting suicide attempt or death following a visit to psychiatric specialty care: A machine learning study using Swedish national registry data”

3.Abstract: Methods and Findings: Given the similarity of mean ages of males and females, it would be more informative if you could please include the % of males and females, and the overall mean age. It would also be helpful to have a breakdown of mental/behavioral disorders represented, as well as percentages for participant ethnicity.

4.Methods: Page 9, line 15: Please specify the significance level used (eg, P<0.05, two-sided) where you describe the Hanley and McNeil test used to derive a p value.

5.Methods: Page 9, line 19: Please change “algorisms” to “algorithms”

6.Methods: Page 9, lines 22-24: Thank you for noting your analyses were preplanned, and that there is no analysis plan documented. “The analyses were planned in April 2019 by the project team, and then revised after further discussions between QC, HL, and SF in September 2019, but we did not publish a study protocol.” Please make sure that the Methods section transparently describes which analyses were planned, and when/why any data-driven changes to analyses took place. Any changes in the analysis-- including those made in response to peer review comments-- should be identified as such in the Methods section of the paper, with rationale.

7.Results, Page 11, Line 24: Please present the exact p value or p <0.001

8.Discussion, Page 14, Lines 1-4: Please clarify the following sentence: “Novel and previously untested data may open opportunity opportunities for deep learning analytics to improve prediction of suicidal behavior through identifying highly complex data pattern [35, 36]. especially if the complexity is beyond human comprehension for the time being.”

9.TRIPOD Checklist: Please refer to sections rather than page numbers in the Checklist. For example, for “Title” please refer to the “Title Page” and for “Abstract” please refer to Abstract as the section.

10. Table 3: Please note in the legend the meaning of “...” entries in the table in the legend.

Comments from Reviewers:

Reviewer #3: Thank you for your explanation on the calibration plots. eTable 9 that was added to the supplementary information was very helpful in understanding and interpreting the plots. It would strengthen your understanding, as well as readers understanding, of the models, if you added information about the number of unique people in each row in eTable 9. It is difficult in this setting, but those 31 visits in the bottom row on eTable 9 could be to the same person and all be associated with the *same* suicide attempt. Alternatively they could be visits to the same person (or a small group of patients) with multiple suicide attempts per person or they could represent 31 different visits by 31 visit people. My guess is that this is for a small group of people (maybe only one person) that is in this highest threshold. This makes me really want to see what the performance of this model would be in a different data set, set this tight performance as the highest risk could be overfitting or it could be a model that is able to identify highest risk individuals well.

[LINK]

---

## [Editor Report · Decision Letter 5]

8 Oct 2020

Dear Dr. Chen, 

On behalf of my colleagues and the academic editor, Dr. Alexander C. Tsai, I am delighted to inform you that your manuscript entitled "Predicting suicide attempt or suicide death following a visit to psychiatric specialty care: A machine learning study using Swedish national registry data" (PMEDICINE-D-20-00084R5) has been accepted for publication in PLOS Medicine. 

PRODUCTION PROCESS

Before publication you will see the copyedited word document (within 5 busines days) and a PDF proof shortly after that. The copyeditor will be in touch shortly before sending you the copyedited Word document. We will make some revisions at copyediting stage to conform to our general style, and for clarification. When you receive this version you should check and revise it very carefully, including figures, tables, references, and supporting information, because corrections at the next stage (proofs) will be strictly limited to (1) errors in author names or affiliations, (2) errors of scientific fact that would cause misunderstandings to readers, and (3) printer's (introduced) errors. Please return the copyedited file within 2 business days in order to ensure timely delivery of the PDF proof. 

If you are likely to be away when either this document or the proof is sent, please ensure we have contact information of a second person, as we will need you to respond quickly at each point. Given the disruptions resulting from the ongoing COVID-19 pandemic, there may be delays in the production process. We apologise in advance for any inconvenience caused and will do our best to minimize impact as far as possible.

PRESS

PROFILE INFORMATION

Thank you again for submitting the manuscript to PLOS Medicine. We look forward to publishing it. 

Best wishes, 

Caitlin Moyer, Ph.D.

Associate Editor 

PLOS Medicine

plosmedicine.org